# Measurement of the combined quantum and electrochemical capacitance of a carbon nanotube

Jinfeng Li[1] & Peter J. Burke[2,3,4,5]

The nature of the electronic interface between a nanotube and solvated ions in a liquid electrolyte is governed by two distinct physical phenomena: quantum and chemical. The quantum component arises from the sharply varying electronic density of states and the chemical component arises from ion screening and diffusion. Here, using an integrated on-chip shield technology, we measure the capacitance of one to a few nanotubes quantitatively as a function of both bias potential (from −0.7 V to 0.3 V) and ionic concentration (from 10 mM to 1 M KCl) at room temperature. We determine the relative contributions of the quantum and electrochemical capacitance, and confirm the measurements with theoretical models. This represents an important measurement of the quantum effects on capacitance in reduced dimensional systems in contact with liquid electrolytes, an important and emerging theme in the interface between nanotechnology, energy, and life.

[1] Department of Physics and Astronomy, University of California, Irvine, CA 92697, USA. [2] Department of Chemical Engineering and Materials Science, University of California, Irvine, CA 92697, USA. [3] Department of Biomedical Engineering, University of California, Irvine, CA 92697, USA. [4] Chemical and Materials Physics program, University of California, Irvine, CA 92697, USA. [5] Department of Electrical Engineering and Computer Science, University of California, Irvine, CA 92697, USA. Correspondence and requests for materials should be addressed to P.J.B. (email: pburke@uci.edu)

One of the most fundamental properties of the interface between matter and liquid is the capacitance of that interface. The capacitance governs the ability to carry charge, and therefore determines the final fate of any electrical currents flowing through that interface. In biological systems, the capacitance directly affects the speed of propagation of the action potential along neurons[1], and plays an important role in many other bio-electronic phenomenon, including the beating of a cardiomycyte[2], as well as the creation and consumption of energy in organelles such as chloroplasts for photosynthesis and mitochondria for oxphos and ATP synthesis[3]. The capacitance also plays an important role in energy storage technologies such as batteries and supercapacitors[4–6], and governs the behavior of numerous electrochemical sensors[7].

Until this work, the size of typical electrodes has been macroscopic compared to the de Broglie wavelength of the electrons in the solid. With the modern advent of nanomaterials, that has changed, and reduced dimensional materials such as 2d materials (graphene and beyond graphene), 1d materials (nanoribbons, nanotubes, nanowires), 0d materials (quantum dots) give rise to a whole class of electronic devices in which the size of the structure becomes comparable to the electron wavelength. Just as Bohr surmised at the dawn of quantum mechanics that the wave function as an electron circles an atom must return to its original value (hence the allowed wavelengths and energies in an atom are quantized), this gives rise to quantization of energy levels within any solid whose size approaches the electron wavelength in any dimension.

What is the effect of this on the capacitance between a reduced dimensional system and a liquid electrolyte? While this has been addressed in theory, in practice for $d < 2$, it has been impossible (until now) to answer for one very important practical reason: It is almost immeasurably small. The quantum capacitance of a nanowire/nanotube is generally of order 100 aF $\mu m^{-1}$. In dry systems, it is possible to isolate the nanodevice from the system[8] but in an electrolyte system, the electrolyte contacts everything and creates a stray overlap capacitance between contact electrodes and the nanodevice, which swamps the signal that is being measured.

At the same time, the "classic" Debye layer capacitance deviates from textbook behavior when the radius of curvature of the electrode becomes comparable to the solvated ion radius, also of the order of 1 nm. This can occur in one of two topologies: Nano-caves, and nano-electrodes. In a nano-cave, the cave size becomes small in a porous material giving non-trivial capacitance, changing the behavior by up to 3 times the classic calculation. This was discovered experimentally and only later explained by electrochemical simulations[9–12]. In nano-electrodes, the electrode protrudes into the liquid, and the predicted behavior deviates substantially from the classic one[13–16]. The results of both of these effects (quantum and electrochemical) give rise to a new regime of electrochemical behavior for nanosystems, qualitatively different from both by new well-studied dry nano-systems and classic large area electrochemical systems.

In this work, in order to resolve the small capacitance (of order 100 aF) above the background stray capacitance (of order 100 pF), we design, develop, and implement an integrated, on-chip shield. With this technique, we measure the capacitance between a 1d material and an electrolyte. With this system, we measure the capacitance of one to a few single-walled carbon nanotubes (SWNTs) quantitatively as a function of both bias potential (from −0.7 V to 0.3 V) and ionic concentration (from 10 mM to 1 M KCl) at room temperature. The corresponding capacitance in this case consists of two types of capacitance in series: a quantum component arising from the electronic density of states and an electrochemical component arising from the ion screening and

diffusion. By varying the electrolyte concentration, we determine the relative contributions of the quantum and electrochemical capacitance. This technique, proven in concept for the case of carbon nanotubes, is applicable to a broad class of reduced dimensional devices, including nanowires, nanoribbons, and quantum dots, of any material. Thus, this represents an important measurement of the quantum effects on capacitance in reduced dimensional systems in contact with liquid electrolytes, an important and emerging theme in the interface between nano-technology, energy, and life.

## Results

**Shield concept and device description.** Figure 1 illustrates the integrated on-chip shield concept and design. In the unshielded measurement geometry (Fig. 1a), the measurement of the nanotube-electrolyte capacitance is typically done in parallel with a much larger background parasitic capacitance which must be subtracted off. With the liquid in capacitive contact with the contact pads, this procedure is not feasible, as the parasitic capacitance is typically of order 100 pF and the capacitance to be measured is of order 100 aF, six orders of magnitude smaller. Note that microfluidic confinement of the electrolyte may help to reduce the parasitic capacitance, however, to reach the required level (order of 100 aF), it requires an extremely narrow channel which will generate large streaming potential and significant noise due to the current fluctuation along the channel and the local potential drift at the device gate[17–19]. (In a dry environment, the liquid is not present so this procedure is feasible[8,20]). In order to mitigate the effects of the unwanted parasitic capacitance on the measurement, we implemented a "shield" which shunts all parasitic capacitive current to ground, while still allowing the capacitive current through the device-electrolyte interface to flow into the current amplifier to be measured (Fig. 1b). The Si/SiO₂ substrate is used as a bottom shield to further eliminate the background parasitic capacitance. Thus, the top and bottom shields together form a sandwich-like structure that effectively shields out the parasitic current as well as noise from the environment.

The detail of the device fabrication is in the Methods section. Briefly, a pair of source-drain electrodes are first fabricated on top of a highly doped low-resistivity silicon wafer with a 300 nm oxide layer in between, using standard photo and e-beam lithography followed by metal evaporation and lift-off processes. A three-layer structure (dielectric/metal/dielectric layers) is then fabricated layer-by-layer on top of the source-drain electrodes, covering everywhere except for a small window opened in the center of the device. This window is made in a cross shape with each edge having a length of 8 μm. It allows the front tips of the electrodes to be exposed and form contact with liquid solutions. SU-8 is chosen as the dielectric material, which is spin-coated to have ~1 μm in thickness and patterned using photolithography. For the metal layer, 50 nm gold is deposited with 30 nm of Ti underneath to increase adherence to the dielectric layer. After the three-layer structure is made, the device is brought under oxygen plasma to etch off any photoresist residue on the contact electrodes. Next, nanotubes are deposited across the source-drain contact electrodes using dielectrophoresis (DEP) method[21–26]. We drop a purified SWNT-suspended solution (from NanoIntegris) into the central window and use DEP force to attract the SWNTs to align and attach onto the contact electrode tips. An anneal (160 °C, 5 min) is used afterward to enhance the electric contact between the electrode tips and the SWNTs. Finally, the device is spin-coated with PMMA as the final passivation layer, and a 600 nm width channel is opened using e-beam lithography to expose the SWNTs and allow the aqueous solution to come in

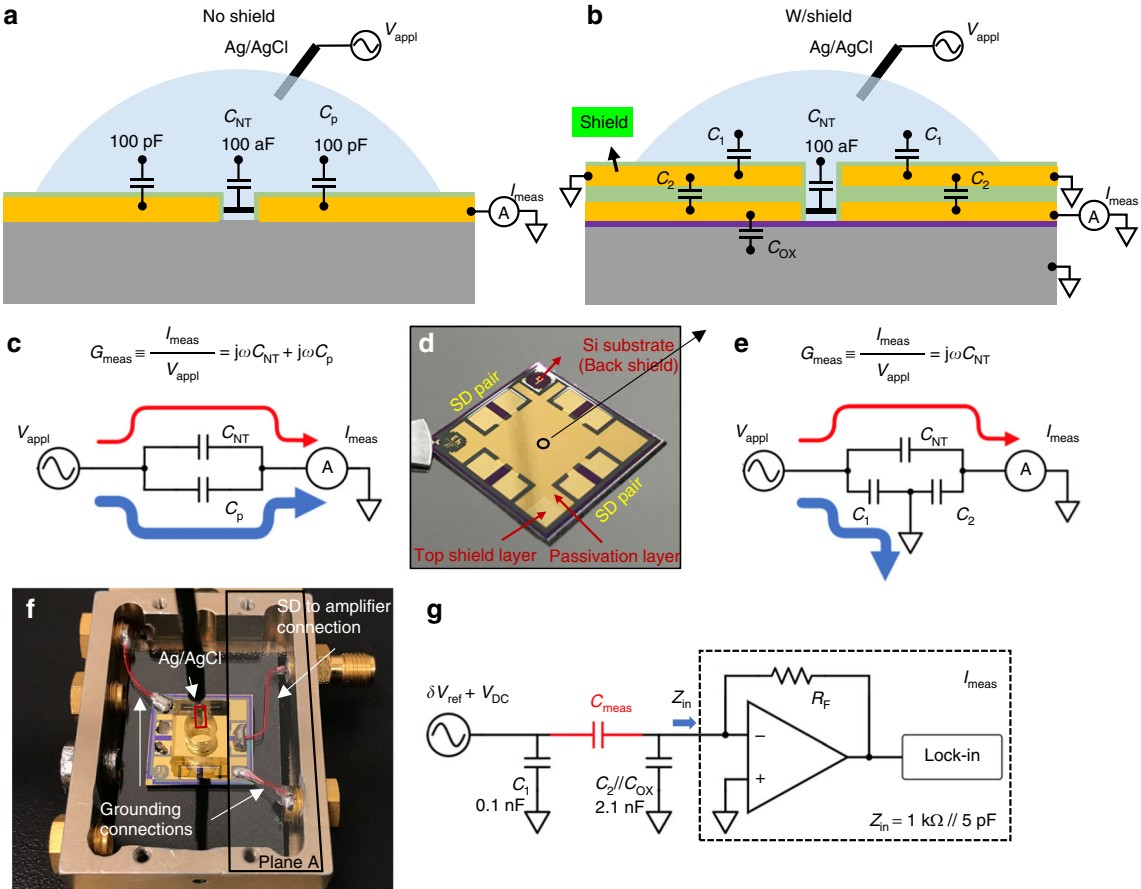

**Fig. 1** Liquid-gated nanotube devices with an integrated on-chip shield. **a, c** Schematic diagrams and corresponding circuits of a nanotube device without integrated shield, versus **b, e** with integrated shield. For non-shielding case, the measured current between the liquid and the contact electrodes contains parasitic current (blue arrow line in subfigure **c**) that is ~6 orders of magnitude larger than the current passing through the nanotube (red arrow line), which easily swamps the current signal of interest and make it too difficult to measure. For the shielding case, the parasitic current (blue arrow line in subfigure **e**) is directed to the ground instead of the current meter. **d** Optical image of a device chip. The source-drain electrodes are patterned underneath the shielding layer and form contact with nanotubes in the center of the chip. **f** Wire connections of a device sitting inside a Faraday box. Copper wires between the device and the SMA connectors are highlighted with red color. A PDMS chamber is placed on top of the device as a liquid reservoir. The Ag/AgCl reference electrode is brought in the reservoir by a coaxial cable. Plane A is covered with a grounded metal plate during the measurement to further eliminate the parasitic current. **g** Circuit model of the device and the measurement units. Currents that pass through the parasitic capacitances ($C_1$, $C_2$, and $C_{ox}$) are directed to the ground. Currents that pass through the SWNT-electrolyte interface are input to the pre-amplifier and the lock-in amplifier, to characterize the corresponding capacitance $C_{meas}$. The input impedance of the pre-amplifier is 1 kΩ // 5 pF

and form contact. Due to the hydrophobic properties of the small central chamber, bubbles are easy to form and prevent the contact. Therefore, we always wet the device with alcohol before adding any aqueous solution. In the same batch of devices, one random picked device will omit and only omit the SWNT deposition step to behave as a control device, which allows us to estimate the remaining parasitic capacitance. We observed the parasitic capacitance for a control device is ~1 fF, and the measured capacitance of the control device has no dependence on both the back and the top gate voltage.

**DC characterization and device yield**. We first characterize the nanotube devices by measuring the gate-dependent source-drain conductance (see Fig. 2). A voltage bias (100 mV) is applied between the source-drain electrode and the current is monitored at the source electrode to quantify the conductance of the nanotubes. The nanotube is first back-gated by the silicon substrate between −10 V and +10 V. After the back-gate measurement, a droplet of electrolyte (100 mM KCl) is placed on the center of the device, and a top-gate voltage (between −0.7 V and 0.4 V) is applied to the electrolyte solution using an Ag/AgCl

reference electrode. Fig. 2a shows the source-drain current as a function of the back-gate voltage, and Fig. 2b shows that as a function of the top-liquid-gate voltage. In the switching range of both gates, exponential dependence is observed, suggesting semiconducting SWNTs are formed across the source-drain electrode tips. The slope of the subthreshold in the back-gate measurement is ~800 mV per decade, while in the liquid-gate measurement, due to the strong coupling of ions, the slope is large and about 60–90 mV per decade, agreeing with others' works[27,28]. The SWNTs behave in the p-type conductive region at negative gate voltage and no n-type conductance is observed in our measurement range. The back-gate measurement in a dry environment shows a large hysteresis phenomenon[29], however in the liquid-gate case, the hysteresis phenomenon is very minor. The transport characteristics for multiple devices are shown in the inset of Fig. 2a; the different ON-state current and threshold voltage suggests the different number and doping of SWNTs formed across the source-drain electrode tips. The liquid-gate current is shown in the inset of Fig. 2b, which is minor and mainly capacitive current rather than charge transfer current at the SWNT's surface. (Note that the transport measurement

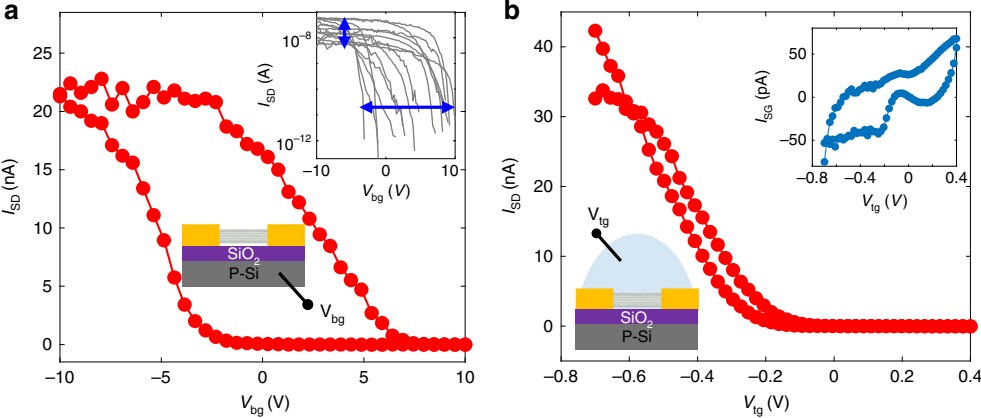

**Fig. 2** Gate-dependent source-drain current of the nanotube devices. **a** Source-drain current as a function of the back-gate voltage before applying the liquid-gate. The source-drain bias voltage is 100 mV, and the back-gate voltage sweeps between −10 V and 10 V at 3 V s⁻¹. The inset shows the source-drain current on a logarithmic scale for different devices. **b** Source-drain current as a function of the top liquid-gate voltage. The source-drain bias voltage is 100 mV, and the liquid-gate voltage sweeps between −0.7 V and 0.4 V at 0.1 V s⁻¹. The inset shows the capacitive gate current before applying the shield connections

was done prior to the shield connections and capacitance measurement).

The two-stage DC transport measurements (the back-gated and the liquid-gated) are also used as a verification of the devices prior to the electrochemical capacitance measurements. 42 devices were tested, in which 18 devices showed on/off switch when back-gate was applied (the inset of Fig. 2a), which confirmed semiconducting nanotubes had been successfully deposited across the source-drain electrodes and formed a good electrical contact. Next, in the liquid-gate test, 4 out of the 18 devices showed on/off switching, which verified the liquid solution had contacted the nanotube directly, with no bubbles or photoresist residue in between. These 4 devices were then characterized using the electrochemical capacitance measurement technique, discussed below. Although measurement on a large number of devices can provide better statistical analysis, capacitance measurement on these 4 devices has shown consistency with each other both on the overall shape of the capacitance and the magnitude of value.

**Capacitance measurement and theoretical interpretation.** Once the DC transport characteristics confirm the presence of a well-behaved semiconducting nanotube in the device and the liquid solution has directly contacted the nanotubes, an AC perturbation voltage is superimposed on top of the DC liquid-gate to measure the capacitance as follows (Fig. 1). An AC voltage (1 kHz, 50 mV in RMS) is applied to the electrolyte using an Ag/AgCl reference electrode, and the source and drain of the nanotube are wired together and connected to a virtual ground which measures the corresponding AC current between the electrolyte and the nanotube. The AC voltage on the electrolyte causes a capacitive current to flow into the nanotube and into the virtual ground; the measured capacitance is then determined by $V_{AC}/I_{meas} = 1/j\omega C_{meas}$. In practice, there are additions to this, such as the ohmic contact resistance of the SWNT to gold, the nanotube resistance, the electrolyte resistance, and a possible faradaic current from the nanotube to the electrolyte. These resistive components contribute negligible measured current in comparison to the capacitive current of the nanotube-to-electrolyte. The relative ratio between the imaginary and real part of the impedance is ~3, and the corresponding phase shift mainly comes from the imperfect double layer capacitor, as discussed in detail in Supplementary Note 1. A low noise pre-

amplifier (FEMTO DLPCA-200, with input noise current 13 fA per √Hz) followed by a lock-in amplifier (Stanford Research SR830) is used for the current measurement. The top and bottom shield layers are grounded; the source-drain electrodes are connected to the input of the transimpedance pre-amplifier. Most of the parasitic current is terminated at the grounded shields, only the current that passes through the liquid-SWNT interface finally gets carried out to the measurement electronics. To further eliminate the parasitic current and environmental noise, a grounded aluminum foil is used to cover the exposed source-drain electrode areas (plane A in Fig. 1f) such as the soldering points and the current collection wires. After proper shielding, the parasitic capacitance measured on a control device is decreased to ~1 fF. This is essential to make the capacitance measurement possible for individual SWNTs exposed to conductive liquid.

We first present our raw data on the measured capacitance as a function of gate voltage and electrolyte concentration. Fig. 3 is a representative trace from one of the devices. It shows the measured capacitance of nanotubes as a function of the liquid-gate voltage at different electrolyte concentrations (10 mM, 100 mM, and 1 M KCl). It shows a clear threshold around a gate voltage of −0.2 V, with a dramatic vanishing of $C_{tot}$ when the gate voltage increases in the positive direction above the threshold voltage, and a point of inflection and then increase when the gate voltage decreases below the threshold volage. Multiple cyclic sweeps are shown in Fig. 3a and show consistecy from sweep to sweep within 10%, with little hysteresis. A detailed analysis shows that all 4 devices demonstrated this qualitative behavior (see below). In some of our measurements (see below), we observed one or two subpeaks in the uptrends of the capacitance curve, which we believe are related to the van-Hove singularities at the rise of hole sub-bands. However, the subpeaks are not always observed. At this point, the reason peaks are not always observed is not clearly known. Some possibilities include the different diameters and arrangements of the nanotubes between the contact electrodes and the possible existence of surface residues that work as a third capacitor. Fig. 3b also shows the on-state capacitance as a function of the ionic concentration. The capacitance goes up as the ionic concentration increases. This is consistent with the qualitative expectation of increased capacitance due to decreased DeBye screening length.

Now that we have presented the raw data, we turn to the theoretical interpretation. Our model is summarized in Fig. 4c.

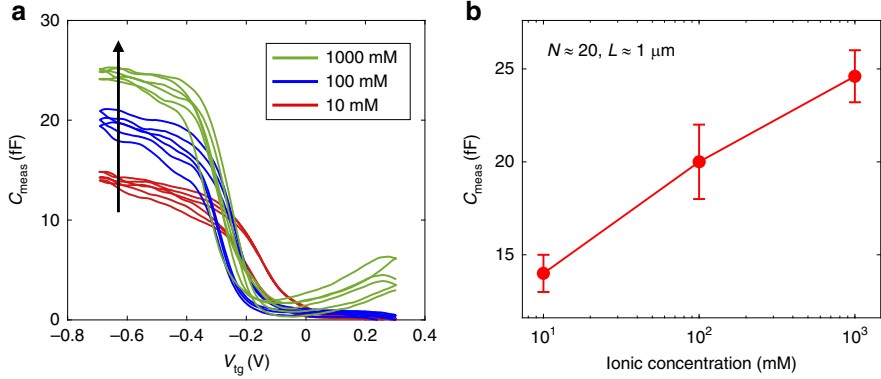

**Fig. 3** Capacitance measurement. **a** The measured nanotube capacitance as a function of the liquid-gate voltage at three different ionic concentration (10 mM, 100 mM, 1 M). Three cyclic scans at 100 mV s$^{-1}$ are shown for each concentration. **b** The measured total capacitance as a function of ionic concentration at a fixed gate potential −0.7 V, with confidence intervals

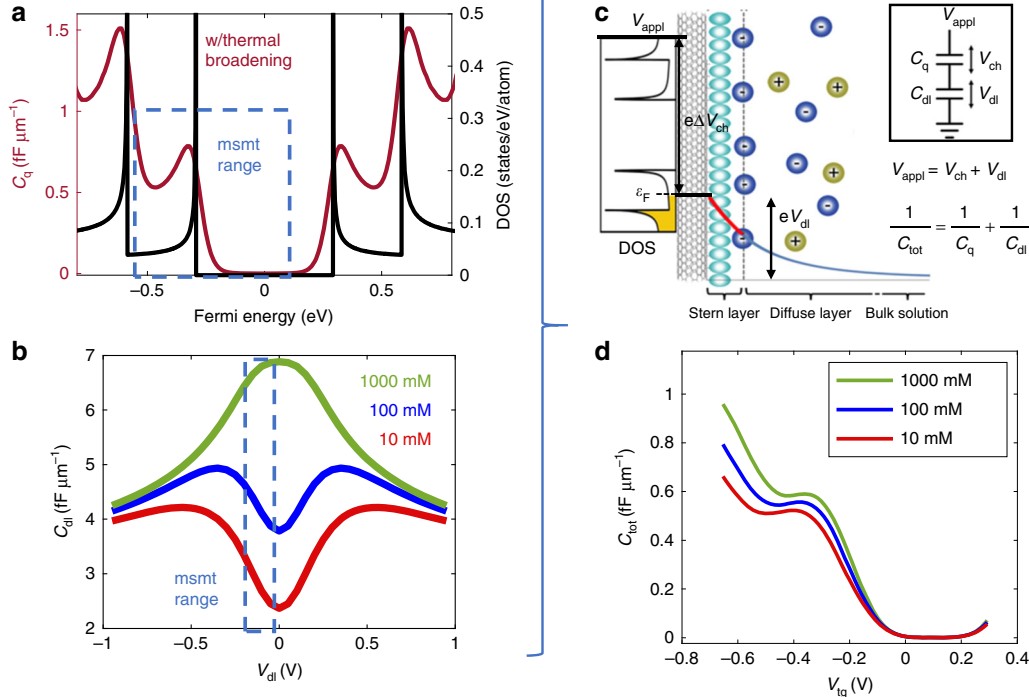

**Fig. 4** Calculated quantum capacitance and double layer capacitance. **a** The calculated density of states (DOS) of a SWNT with a diameter 1.2 nm (black curve) and the resulting quantum capacitance (red curve) at room temperature. **b** Calculated double layer capacitance of a nanocylinder with diameter 1.2 nm, immersed in KCl aqueous solution with three different concentrations (10 mM, 100 mM, 1 M). **c** Schematic of the double layer structure with the state alignment between the nanotube and the solution. The applied voltage drops in part as the change of chemical potential of the nanotube and then decays exponentially in the Stern and diffuse layer in the solution. **d** The calculated total capacitance for three different ionic concentrations

The applied voltage is divided between the two capacitors: the quantum one and the double layer one, and this ratio will depend on the value of the capacitances, which depends on the electrolyte concentration through the dependence of $C_{dl}$ on the electrolyte concentration. As discussed in our previous paper[16], $C_{dl}$ vs. $V_{dl}$ can peak or trough at the origin depending on the concentration. Fig. 4b shows this theoretical dependence for three concentrations (10 mM, 100 mM, 1 M). This can be combined with the quantum capacitance calculations to predict the dependence of the total capacitance on electrolyte concentration. As can be seen in Fig. 4b, the overall effect of the increasing concentration in our voltage window is to increase $C_{dl}$. This would increase the overall capacitance, and this trend is indeed observed in the data, as shown in Fig. 3. Whereas the quantum capacitance depends only on the internal quantum structure of the nanotube, the

electrochemical capacitance depends strongly on the molarity of the solution. This gives us an experimental "knob" to determine the relative contributions of the two capacitances to the total capacitance quantitatively.

To normalize the electrochemical capacitance of nanotubes with per unit length, one needs to know the total length of the nanotubes that contributes to the measured signal and divide the measured capacitance by the total length. However, due to the nature of the DEP method, the total number and length of nanotubes are not known exactly. Here, we provide an estimation of the total length based on SEM imaging. Since some tubes may not be electrically connected, and some are bundled, hidden or hardly visible under SEM, the length determined with this method is only an approximation. The device corresponding to the measurement in Fig. 3 has ~20 nanotubes counted and the

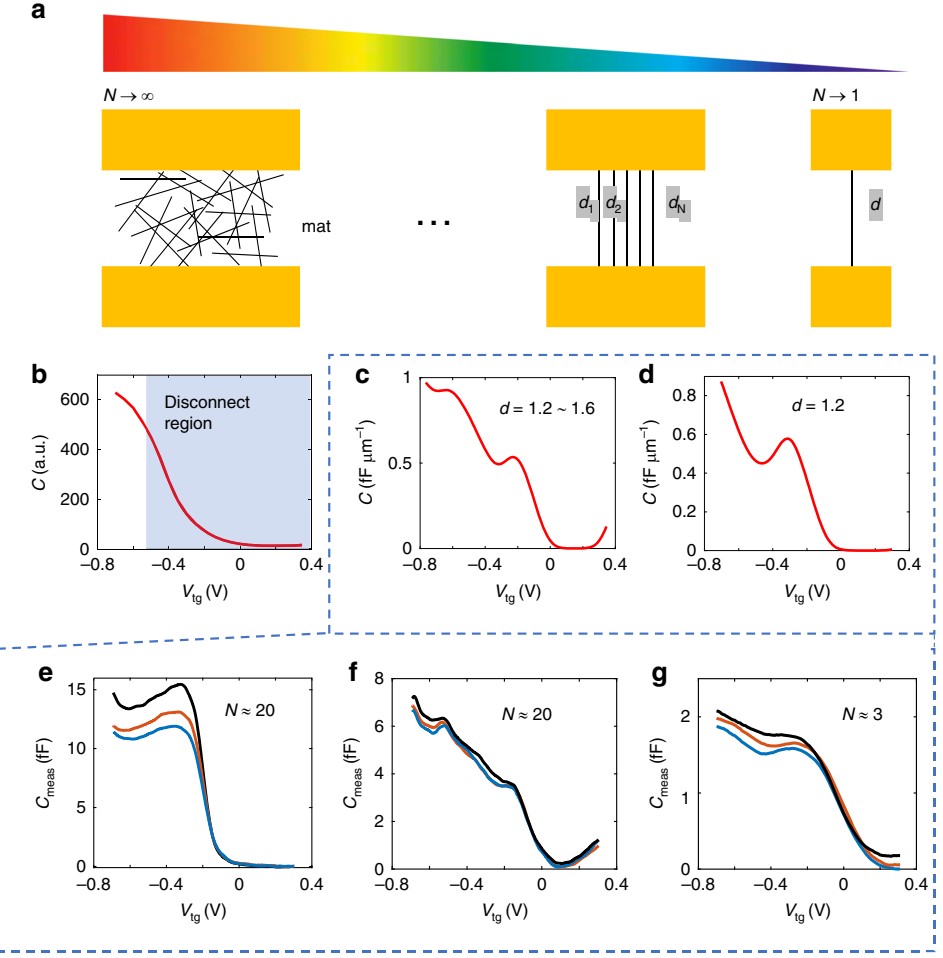

**Fig. 5** Effect of nanotube arrangement. **a** Diagrams of nanotube devices with different tube diameter, quantity, and arrangement. **b**–**d** The corresponding capacitance curves. The mat case was measured by us in ref. [16]. In that case, the source drain resistance was too high between the electrodes and the nanotubes in the center of the mat, and so we could not measure the capacitance in the off region, labeled as "disconnect region" in **b**. In contrast, in this work, we are able to measure the capacitance through the entire voltage region, including both above and below the threshold voltage, as show in **e**, **f** and **g**: the measured capacitance curves for different devices

average length of the tubes is ~1 μm. If we divide the measured capacitance by the estimated total length, the averaged capacitance per length is ~1 fF μm$^{-1}$, consistent with our theoretical model presented in Fig. 4. The other devices have nanotubes counted as ~3, 10, 20, and their capacitance data are plotted in the bottom panel of Fig. 5. Although the overall "step" shape remains among different measurements, the details of the capacitance curves vary slightly, especially the number and position of the subpeaks. Later in this paper, we provide an explanation based on the different arrangements of nanotubes. A different approach (such as CVD-grown nanotubes) rather than DEP can provide better control to the nanotube arrangement, hence a higher certainty when comparing the measurement with theory. However, the overall shape of the capacitance between a few nanotubes and an electrolyte solution, and its approximate value have been measured in this work with the integrated on-chip shield, which can be used as a basis for the future measurements and applications.

We turn to a more detailed theoretical analysis of the non-idealities of our measurements (e.g., single tube vs. multiple heterogenous tubes). In our recent work[16], we treated the extreme case of a large number of tubes. This paper extends that work into the qualitative regime of only a few tubes, giving insight into the behavior of single tubes. We first model a single nanotube case,

then extend it to the case of multiple mixed nanotubes. The total capacitance $C_{tot}$ is a combination of the intrinsic quantum capacitance and the electric double layer capacitance of a carbon nanotube. The quantum capacitance is proportional to the DOS, which is sharply dependent on the Fermi energy. At room temperature, this dependence is thermally broadened. Additional measurement broadening is due to the finite applied AC voltage swing and the variation of the tube diameters, which will be discussed later. Fig. 4a shows the DOS and thermally broadened quantum capacitance of a SWNT with 1.2 nm diameter as functions of the Fermi energy. The DOS of a SWNT is calculated based on zone-folding of graphene's band structure[8,30]. In addition, the electrochemical capacitance depends on the electrolyte-nanotube potential drop. The theory of this for 1d conductors is non-trivial and was discussed by us in ref. [16]. and others[14,15]. Fig. 4b shows the predicted electrochemical double layer capacitance as a function of the double layer potential $V_{dl}$ in the solution. The voltage drop and Fermi energy together sum to the applied voltage, and the two capacitors add in series, as shown in Fig. 4c. Since we use an aqueous electrolyte, we restrict the applied voltage between −0.7 V and 0.3 V to avoid electrolysis. This restricts the measurement range of the Fermi energy $E_F$ and the double layer voltage $V_{dl}$ to be the box regions shown in Fig. 4a, b.

Figure 4d shows the combined theoretical capacitance for three different concentrations. The main predicted features of this model agree with the measured data: (1) The total capacitance is dominated by the quantum capacitance, due to its smaller capacitance value compared to the double layer one. (2) The capacitance vanishes near the origin due to the quantum capacitance component vanishing in the bandgap. (3) As the gate voltage is reduced below the threshold voltage, the capacitance sharply rises, due to the arising of the first sub-band in the 1d density of states. (4) The double layer capacitance increases with the ionic concentration, which causes the total capacitance to be dependent on the ionic concentration. (5) During the rise of the capacitance, there is a slowdown in the uptrend due to the slow decrease of the DOS after the reach of the first sub-band. The slowdown of the capacitance going up, together with the arising of the second sub-bands caused a peak in the capacitane, arising from the van Hove singularity in the density of states. All of our measurements showed characteristics #1–4, but only some showed #5. (Note that our recent work on multiple tubes in ref. [16] was not able to access the gap region due to the large source-drain spacing and hence large resistance in the off case in that work.) At this point, the reason the van Hove peaks are sometimes smeared out is not clearly known. Some possibilities include the different diameters and arrangements of the nanotubes between the contact electrodes and the possible existence of surface residues that work as a third capacitor, hence smearing out the van Hove peak.

We next discuss the detailed calculation that led to the capacitance curves in Fig. 4. We calculate each capacitance component (quantum and electrochemical) separately and then combine them together in series to calculate the total capacitance as a function of the total applied voltage.

$$C_{\text{tot}}^{-1} = C_{\text{q}}^{-1} + C_{\text{dl}}^{-1} \tag{1}$$

The quantum capacitance at a given chemical potential $V_{\text{ch}}$ is given by [8,30,31],

$$C_{\text{q}}(V_{\text{ch}}) = \int dE \cdot F_{\text{T}}(E - eV_{\text{ch}}) \cdot C_{\text{q}}^0 \sum_{j=-3}^{3} \left(1 - \left(\frac{E_j}{E}\right)^2\right)^{-1/2}$$
$$E_j = \hbar v_{\text{F}} \frac{2j}{3d}, C_{\text{q}}^0 = \frac{4e^2}{\pi \hbar v_{\text{F}}} \tag{2}$$

where, $F_{\text{T}}(E) = (4k_{\text{B}}T)^{-1} \text{Sech}^2(E/2k_{\text{B}}T)$ is thermal broadening function, $k_{\text{B}}$ is Boltzmann constant, $T$ is temperature, $v_{\text{F}}$ is the Fermi velocity, and we included the first three electron and hole sub-bands. Fig. 4a shows the calculated quantum capacitance as a function of the Fermi energy. Due to the choice of the reference electrode and the doping of the SWNT, the potential of the SWNT at the half-filling state has an offset (0.1–0.2 V) with respect to the Ag/AgCl reference electrode. This offset can be estimated by aligning the threshold of the measured capacitance-voltage curve with the theoretical curve.

The double layer capacitance for a carbon nanotube has been calculated in our previous work, based on a modified Poisson–Boltzmann equation[13,16,32–35],

$$\frac{1}{r}\frac{d}{dr}\left(\varepsilon_{\text{r}}\varepsilon_0 r \frac{d}{dr}\right)\varphi = \frac{2\rho_{\text{q}}\sinh\left(\alpha \cdot \frac{q\varphi}{k_{\text{B}}T}\right)}{1 + 2\nu \sinh^2\left(\alpha \cdot \frac{q\varphi}{2k_{\text{B}}T}\right)} \tag{3}$$

where $\varphi$ is the electric potential distribution along the radial direction of a nanotube in solution, $\rho_{\text{q}}$ is the space charge density in the bulk solution, $\alpha$ is the correlation parameter, $q$ is the charge of the ions, and $\nu$ is the packing parameter. Together with the appropriate boundary conditions listed in our previous work[16], we can calculate the potential distribution in the contact solution.

From the potential distribution, the double layer capacitance can be determined,

$$C_{\text{dl}}(V_{\text{dl}}, c_0) = \frac{dQ}{dV_{\text{dl}}} = -\varepsilon_{\text{r}}\varepsilon_0 \frac{d}{dV_{\text{dl}}} \iint_{r=r_{\text{H}}} \nabla\varphi \cdot \mathbf{dS} \tag{4}$$

Figure 4b shows the calculated double layer capacitance of a nano-cylindrical electrode with 1.2 nm diameter, in three different KCl aqueous solution (10 mM, 100 mM, and 1 M).

With the two types of capacitance in series, the applied liquid-gate potential $V_{\text{appl}}$ splits into two parts: the chemical part $V_{\text{ch}}$ over the quantum capacitance $C_{\text{q}}$, and the electrostatic part $V_{\text{dl}}$ over the double layer capacitance $C_{\text{dl}}$. The ratio between the two capacitances is fully determined by the ratio of the potential drop between the two,

$$\frac{C_{\text{q}}}{C_{\text{dl}}} = \frac{V_{\text{dl}}}{V_{\text{ch}}} \tag{5}$$

Combining the equations 1–5, we can calculate the total capacitance as a function of the applied gate potential.

**The effect of nanotube arrangement**. We can now compare the different extremes of single tube vs. many tubes, shown in Fig. 5. In prior work[16], we measured the ensemble capacitance for millions of nanotubes in a mat. In this work, we provide a bridge to that work by measuring a few nanotubes using our integrated on-chip shield. The bottom panel of Fig. 5 shows the capacitance curve for different devices. They all show an increase of capacitance when the liquid-gate voltage goes more negative. However, the detailed shape of the capacitance varies among devices. For example, in Fig. 5f there are two peaks on the uptrend, whiles in the other two measurements (Fig. 5e, g) there is only one peak on the uptrend. In addition, the uptrend shown in Fig. 5e is sharper than the others. These differences are very likely caused by different arrangements of nanotubes between the contact electrodes (Fig. 5a). For a single nanotube device, the arrangement is the simplest one. Fig. 5d shows the theoretical curve for a single nanotube, showing a similar shape with the measurement in Fig. 5g. For devices that contain a mixture of nanotubes with different diameters, the misaligned sub-bands will overlap and result in a shift-averaged capacitance curve, which can be calculated by,

$$C_{\text{q}}(V_{\text{ch}}) = \int dd \cdot f(d) \cdot C_{\text{q}}(V_{\text{ch}}, d) \tag{6}$$

where $f(d)$ is the probability density function for the distribution of the tube diameters. Fig. 5c shows the averaged capacitance of nanotubes aligned in parallel with diameter between 1.2 nm and 1.6 nm. The theoretical curve agrees with the measured curve in Fig. 5f, regarding the shapes and bump positions at the first two singularities. In addition to the mixture effect, the crossing arrangement between nanotubes also plays a role in the measured capacitance. Since one nanotube turning OFF can potentially cause the adjacent nanotubes disconnected from the measurement circuit. The capacitance curve is expected to decrease sharply in the disconnection region. In the measurement shown in Fig. 5e, we see a sharper slope than other devices, which may be caused by the disconnection effect. Indeed, the corresponding device has more nanotubes crossing each other compared to other devices as overserved under SEM.

## Discussion

Taken collectively, these measurements demonstrate the net effect of quantum capacitance and atomic scale radius of curvature for one, a few, to a macroscopic number of nanotubes in contact with an electrolyte. As such, it provides an experimental foundation

backed by detailed electrochemical calculations and quantum theory that covers the general case of any number of nanotubes in contact with liquid. Although we have shown the quantum and electrochemical capacitance of a carbon nanotube as a 1d wire, this technique is applicable to a broad class of materials, including nanoribbons, quantum dots, nanowires, and any other small capacitance structures in contact with liquid solution, which is an important and emerging theme in the interface between nano-technology, energy, and life.

## Methods

**Device fabrication.** Carbon nanotube FET devices with integrated on-chip shield were fabricated using the process flow shown in Supplementary Fig. 1. Step 1: A highly p-doped silicon wafer with a 300 nm thermal oxide layer was cleaned using Remover PG, followed by IPA rinse, air dry and baked at 150 °C for 10 mins. Step 2: A layers of photoresist (PMGI SF6) was spin-coated (3500 rpm, 45 s) onto the wafer surface followed by a soft-bake at 170 °C for 5 mins. Another layer of photoresist (Shipley S1827) was spin-coated on top of SF6 layer using the same spin speed and duration, followed by a soft-bake at 115 °C for 90 s. The coated wafer was then brought in contact with the photomask and exposed to UV light (10 mW cm$^{-2}$ @ 365 nm, 20 s) using Karl Suss MA6 Mask Aligner. The exposed photoresist was developed for 60 s using Microposit MF-319, rinsed with DI water and air dried. Ti (10 nm)/Au (50 nm) were deposited by e-beam evaporation, followed by a lift-off process using Remover PG for ~10 h. The source-drain base electrodes were made. Step 3: A layer of e-beam resist (MicroChem PMMA A6) was spin-coated (3500 rpm, 45 s) onto the base electrodes, and then baked at 180 °C for 90 s. Tiny scratches were created near the area under e-beam lithography for focus check. The devices were then aligned under SEM and exposed/patterned by e-beam to form the fine electrode tips (shown in the "top view"). The exposed PMMA was developed for 60 s in 1:3 MIBK:IPA solution, followed by IPA rinse, DI water rinse and air dry. Ti (10 nm)/Au (30 nm) were deposited by e-beam evaporation, followed by a lift-off process using Remover PG for ~10 h. Low power sonication was used for 10 s in the end. The devices were then rinsed by fresh Remover PG followed by DI water rinse and air dry. Now the fine-tip electrodes were made onto the base electrodes. Step 4: The devices were dehydrated on top of a hotplate with the temperature slowly ramping up from room temperature to 180 °C and stayed for 10 mins. After cooling down, the devices were spin-coated (500 rpm, 10 s and then 3500 rpm, 50 s) with a layer of negative photoresist (Micro-Chem SU-8) as the dielectric layer. The coated SU-8 layer was then soft-baked with temperature ramping from 65 °C to 95 °C at 5 °Cmin$^{-1}$, staying at 95 °C for 2 mins before cooling down. The devices were aligned and brought in contact with the photomask and exposed to UV light (10 mW cm$^{-2}$ @ 365 nm) for 9 s. The exposed SU-8 was developed for 2 min with sonication used in the last 30 s and continuing to develop in a fresh developer for 1 more minute with sonication, followed by IPA rinse, DI water rinse and air dry. The patterned SU-8 was then exposed to a UV lamp for 3 min and crosslinked on a hotplate with the temperature slowly ramping up to 200 °C and stayed for 10 mins before cooling down. Now the devices were covered with a dielectric layer with only the electrode tips exposed. Step 5: The devices were cleaned with Remover PG for 3 min followed by DI water rinse, air dry, and bake for 5 min at 160 °C. The same process in step 2 was used here to fabricate the metal shield layer, with some changes: 1. oxygen plasma (100 W 60 s) was used prior to the metal deposition to etch the SU-8 surface lightly and hence increase the adhesion between SU-8 layer and the metal shield layer. 2. A thicker Ti (30 nm) was deposited instead of 10 nm for the same purpose. 3. The time of the lift-off process was shortened to 3 h with the help of shaking (60 rpm) to prevent the dissolution of the SU-8 layer and the peel-off of the metal layer. Step 6: The same process in step 4 was used here to fabricate the passivation layer that protects the shield layer. Step 7: After the three-layer structure (dielectric/shield/passivation layers) was made, the devices were brought under oxygen plasma (100 W 10 mins) to etch off any photoresist residue on the tips of the source-drain electrodes. Then, a diluted ultra-purified nanotube suspension (IsoSol-S100 99.9% purity, Nanointegris Inc) was dropped on top of the devices. An AC electric field (1 MHz, 8 V) was applied between the source and drain electrodes for ~3 s. A single or a few nanotubes were expected to be attracted, aligned and attached across the source-drain electrodes. The devices were rinsed, air dried and then baked (160 °C, 5 mins) to increase the bonding between the contact electrodes and the nanotubes. Step 8: PMMA spin-coating and e-beam lithography were used again here to form the final passivation layer and only expose the middle segment of the nanotubes as the open channel for liquid gating. Step 9: The devices were bonded with a PDMS reservoir to hold liquid. Alcohol treatment was used to prevent bubbles on the hydrophobic surface and then the reservoir was filled with an aqueous electrolyte solution.

**Capacitance measurements.** The electrochemical capacitances of the nanotube devices are measured using a lock-in technique (Fig. 1f, g). A fixed AC voltage $\delta V_{ref}$ (1 kHz, 50 mV in RMS) plus a varied DC voltage $V_{DC}$ (−0.7 V to 0.3 V) is applied to the electrolyte solution through an Ag/AgCl reference electrode. The corresponding current between the solution and the nanotubes is collected at the source-drain

electrodes by the pre-transimpedance-amplifier (FEMTO DLPCA-200, input noise: 13 fA per √Hz). The output of the preamplifier is then connected to the lock-in amplifier (Stanford Research SR830, the time constant is set to 3–10 s with 12–24 dB per oct roll-off) to narrow down the bandwidth and quantify the complex current $I_{meas}$. The measured capacitance is then determined by $1/j\omega C_{meas} = \delta V_{ref} /Im(I_{meas})$. The top shield layer and the bottom substrate are grounded so that the large stray capacitances between the solution and the electrode leads and between the leads and the ground are shielded out of the measurement circuit. Additional shields are applied by covering the current collection wires with aluminum foil. After all the shields, the remaining parasitic capacitance is decreased to ~1 fF.

## Data availability
The data that support the findings of this study are available from the corresponding author upon request.

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

## Acknowledgements

Device fabrication was performed at the Integrated Nanosystems Research Facility (INRF). Oxygen plasma and SEM were performed at the Irvine Materials Research Institute (IMRI). We acknowledge support from the Army Research Office through the ARO-MURI Program, ARO-Core Grants, and DURIP (Contract Nos.: W911NF-11-1-0024, W911NF18-1-0076, W911NF-09-1-0319, and W911NF-11-1-0315), National Institutes of Health (Contract No.: CA182384), and the French American Cultural Exchange (FACE) Partner University Fund program.

## Author contributions

J. Li and P.J.B. conceived the experiments. J. Li fabricated the devices and performed the measurements. J. Li and P.J.B. wrote the manuscript.

## Additional information

**Competing interests:** The authors declare no competing interests.

