## [Peer Review File · Nature Communications]

Reviewers' comments:

Reviewer #1 (Remarks to the Author):

The authors reported the measurement of the quantum capacitance of the interface between semiconducting single-walled carbon nanotube and aqueous electrolyte, with the innovative integration of on-chip shields to minimize the effect of the stray capacitance. They observed the vanishing total capacitance in the gap region and studied the dependence of the capacitance on concentration of the electrolyte. Experimental data fits reasonably well with their proposed theoretical model and calculation.

This work is important and will be of great interest to a broad readership. The data support the conclusions well and the analysis is careful. I think this manuscript is of publication quality, except for two minor issues that the authors should address:

- 1) Details of the device preparation and measurement is mentioned with missing information in the manuscript. There should be a method section to summarize the exact procedures of device fabrication with parameters, and also the measurement setup (gain of preamplifier, time constant of lock-in etc), which could be important for reproducing the results.
- 2) The authors proposed that this work would allow the detection of single electron charging at the interface of a nanoscale 1D detector which could be important for biosensing applications, for example 'measure ions flowing through ion channels one ion at a time, sensing the current through individual transporters, and resolve protein function in real time'. However, it is important to understand that these processes are generally very fast (at $\sim 1\mu\text{s}$ or faster time scale). The design of the measurement in this experiment, although the detailed bandwidth is unclear/not specified, is likely to be limited. If the authors would like to extend their discussion into these applications, they should address this issue with more information and possibly potential solutions.

Overall, I suggest publication of this manuscript.

Reviewer #2 (Remarks to the Author):

Overall, this is a potentially interesting paper on the measurement of the capacitance of single-walled carbon nanotubes (SWNTs) to their surrounding electrolyte. This is an interesting topic with potential relevance to many experiments that use SWNTs as electrochemical elements. If the measurements were well-described and executed appropriately, I would happily support publication in Nature Communications.

Unfortunately, in its current form, I do not feel the paper meets these criteria. I will discuss these below, starting with the most important.

- (1) The most important issue relates to the size of the measured capacitance and its relation to the number of tubes and their length. In the figures, the experimental data is reported as a capacitance per unit length. The reader is certainly left with the impression that this was determined by some experimental process.

From the main paper:

"The measured capacitance is normalized per μm length of nanotubes. Due to the uncertainty of DEP method, different number of nanotubes can be deposited across the source-drain electrodes; 218 we observed the number of attached nanotubes varies from a single one up to a hundred. To normalize the measured capacitance per unit length, we estimate the number and total length of the nanotube shown under SEM and divided the measured capacitance by the total length."

However, the supplementary info paints a different story: Quoting from supp. Info:

"One of the 4 devices was checked under SEM, showing ~ 20 tubes attached to the contact electrodes. Another two has similar capacitance suggesting similar number of tubes. The remaining one is estimated to have less tubes (1~5) due to the smaller capacitance."

This seems to indicate that the procedure described in the text (the capacitance was divided by the length measured under SEM) was not correct – only one sample was measured in the SEM. The others must have been estimated in some other way.

The text is misleading at best. At worst, it is misrepresentation.

(2) Overall, the data is not given the kind of careful discussion and analysis it deserves. The key result, the measured capacitance, is given in the last panels of big figures with lots of theory. It is almost an afterthought. In an experimental paper, the data stands on its own and one must be careful to properly clarify what the data says versus what one's interpretations of the data are. The issue given in (1) is a good example. Other issues in this paper is that it is unclear how many samples are measured (info only in the supp), how reliable the cap/length is (not very, based on (1), and if the authors are clearly claiming to see the peaks in the DOS or just suggestive hints. (The data in Fig 7f shows no such features, for example). Overall, the authors should present the key experimental measurements on their own, summarize what they can and cannot conclude based on the data alone (size of C/L, peak or no peak, etc), and then separately make more detailed comparisons to theory.

(3) The paper is, in my judgement, significantly too long and containing many bits of hyperbole. For example, Fig. 1 is appropriate for the intro slide of a talk but completely unnecessary in an article such as this. The other figures often possess redundant info and could easily be reduced to 4-5 concise figures. As for hyperbole, consider this selection near the end of the paper:

"The DOS and hence capacitance will depend strongly on the Fermi energy, and hence voltage. This is a non-trivial dependence that is fundamentally, profoundly, qualitatively, and quantitatively different from existing electrochemistry textbook knowledge and industry. Therefore, this fundamental scientific study will be the basic foundation for a broad class of industrial challenges facing humanity in the 21st century."

The second sentence, that this is unlike anything in electrochemistry, is overblown. Of course every redox molecule has a much more dramatically structured density of states (i.e. quantum levels) than the nanotube. The nanotube case is interesting, but not something magically special. As for the last sentence in the selection above, it is hard to know exactly what it is saying, but it seems to imply an importance to the work ("this fundamental scientific study will be the basic foundation for a broad class of industrial challenges facing humanity...") much more than is at all reasonable. A more modest, pointed, and appropriate summary would greatly improve the MS.

(4) One small point: The authors do not discuss in this paper the role of the resistance of the SWNT in the measurement. (An issue they did address in their previous experiments on mats of nanotubes.) As they are well aware, the resistance of the tube can also lead to a gap in the measured capacitance. Measurements of the phase of the signal as well as estimates of the relative amplitudes of the real and imaginary impedances would clarify this issue.

Overall, I think this is potentially interesting work. With proper representation of its value and limits, it could be worthy of publication in Nature Communications. But in its present form it is inconsistent, misleading, overlong, and hyperbolic. I do not support publication.

Detailed Response to Reviewer's Comments

Reviewer #1

Reviewer Comments:

The authors reported the measurement of the quantum capacitance of the interface between semiconducting single-walled carbon nanotube and aqueous electrolyte, with the innovative integration of on-chip shields to minimize the effect of the stray capacitance. They observed the vanishing total capacitance in the gap region and studied the dependence of the capacitance on concentration of the electrolyte. Experimental data fits reasonably well with their proposed theoretical model and calculation.

This work is important and will be of great interest to a broad readership. The data support the conclusions well and the analysis is careful. I think this manuscript is of publication quality, except for two minor issues that the authors should address:

1) Details of the device preparation and measurement is mentioned with missing information in the manuscript. There should be a method section to summarize the exact procedures of device fabrication with parameters, and also the measurement setup (gain of preamplifier, time constant of lock-in etc), which could be important for reproducing the results.

Response:

In the revised manuscript, we added a separate Methods section. The details of the device preparation and measurement are included.

The corresponding changes in the revised manuscript are listed below:

Page 21, Line 12 – Page 24, Line 9

Reviewer Comments:

2) The authors proposed that this work would allow the detection of single electron charging at the interface of a nanoscale 1D detector which could be important for biosensing applications, for example 'measure ions flowing through ion channels one ion at a time, sensing the current through individual transporters, and resolve protein function in real time'. However, it is important to understand that these processes are generally very fast (at ~1 μ s or faster time scale). The design of the measurement in this experiment, although the detailed bandwidth is unclear/not specified, is likely to be limited. If the authors would like to extend their discussion into these applications, they should address this issue with more information and possibly potential solutions.

Overall, I suggest publication of this manuscript.

Response:

In the revised manuscript, we deleted the claim about single ion detection. We agree with the reviewer that the measurement bandwidth used in our work (< 0.1Hz) is not suitable for the case of ion channel spike-current study. The on-chip shield can help to minimize the noise in the

related devices, which we plan to explore in future projects. In the revised manuscript, we have deleted this claim to avoid confusion.

Reviewer #2

Reviewer Comments:

Overall, this is a potentially interesting paper on the measurement of the capacitance of single-walled carbon nanotubes (SWNTs) to their surrounding electrolyte. This is an interesting topic with potential relevance to many experiments that use SWNTs as electrochemical elements. If the measurements were well-described and executed appropriately, I would happily support publication in Nature Communications.

Unfortunately, in its current form, I do not feel the paper meets these criteria. I will discuss these below, starting with the most important.

(1) The most important issue relates to the size of the measured capacitance and its relation to the number of tubes and their length. In the figures, the experimental data is reported as a capacitance per unit length. The reader is certainly left with the impression that this was determined by some experimental process.

From the main paper:

"The measured capacitance is normalized per μm length of nanotubes. Due to the uncertainty of DEP method, different number of nanotubes can be deposited across the source-drain electrodes; 218 we observed the number of attached nanotubes varies from a single one up to a hundred. To normalize the measured capacitance per unit length, we estimate the number and total length of the nanotube shown under SEM and divided the measured capacitance by the total length."

However, the supplementary info paints a different story: Quoting from supp. Info:

"One of the 4 devices was checked under SEM, showing ~ 20 tubes attached to the contact electrodes. Another two has similar capacitance suggesting similar number of tubes. The remaining one is estimated to have less tubes (1~5) due to the smaller capacitance."

This seems to indicate that the procedure described in the text (the capacitance was divided by the length measured under SEM) was not correct – only one sample was measured in the SEM. The others must have been estimated in some other way.

The text is misleading at best. At worst, it is misrepresentation.

Response:

In the revised manuscript, we redid all the figures and plotted the capacitance data in its raw form, and separately we discussed the "C/L" normalization and the reliability of the length estimation.

The corresponding changes in the revised manuscript are listed below:

***Page 12, Line 3 – Page 12, Line 19
Page 13, Line 12 – Page 14, Line 7
Figure 3 and Figure 5***

Reviewer Comments:

(2) Overall, the data is not given the kind of careful discussion and analysis it deserves. The key result, the measured capacitance, is given in the last panels of big figures with lots of theory. It is almost an afterthought. In an experimental paper, the data stands on its own and one must be careful to properly clarify what the data says versus what one's interpretations of the data are. The issue given in (1) is a good example. Other issues in this paper is that it is unclear how many samples are measured (info only in the supp), how reliable the cap/length is (not very, based on (1), and if the authors are clearly claiming to see the peaks in the DOS or just suggestive hints. (The data in Fig 7f shows no such features, for example). Overall, the authors should present the key experimental measurements on their own, summarize what they can and cannot conclude based on the data alone (size of C/L, peak or no peak, etc), and then separately make more detailed comparisons to theory.

Response:

We have rewritten the majority of the manuscript to present the measurement data first and then carefully interpreted it based on theory. We redid the figures to accommodate this.

We have rewritten the sample yield part and put it in the main manuscript.

We added discussion about the peak and no peak cases and provide possible reasons for that.

We added a detailed experimental method section in the end of the manuscript before the references.

The corresponding changes in the revised manuscript are listed below:

***Page 9, Line 20 – Page 10, Line 7
Page 16, Line 5 – Page 16, Line 15
Page 19, Line 1 – Page 20, Line 20
Page 21, Line 12 – Page 24, Line 9***

Reviewer Comments:

(3) The paper is, in my judgement, significantly too long and containing many bits of hyperbole. For example, Fig. 1 is appropriate for the intro slide of a talk but completely unnecessary in an article such as this. The other figures often possess redundant info and could easily be reduced to 4-5 concise figures. As for hyperbole, consider this selection near the end of the paper:

"The DOS and hence capacitance will depend strongly on the Fermi energy, and hence voltage. This is a non-trivial dependence that is fundamentally, profoundly, qualitatively, and quantitatively different from existing electrochemistry textbook knowledge and industry. Therefore, this fundamental scientific study will be the basic foundation for a broad class of industrial challenges facing humanity in the 21st century."

The second sentence, that this is unlike anything in electrochemistry, is overblown. Of course every redox molecule has a much more dramatically structured density of states (i.e. quantum levels) than the nanotube. The nanotube case is interesting, but not something magically special. As for the last sentence in the selection above, it is hard to know exactly what it is saying, but it seems to imply an importance to the work ("this fundamental scientific study will be the basic foundation for a broad class of industrial challenges facing humanity...") much more than is at all reasonable. A more modest, pointed, and appropriate summary would greatly improve the MS.

Response:

In the revised manuscript:

We reduced the number of figures from 9 to 5, and combined figure 2 and 5 into one figure (figure 1 in the revised manuscript). We also combined figure 7 and 8, and moved some of the figures into supporting information.

We deleted those hyperboles and softened our claims in the revised manuscript.

Reviewer Comments:

(4) One small point: The authors do not discuss in this paper the role of the resistance of the SWNT in the measurement. (An issue they did address in their previous experiments on mats of nanotubes.) As they are well aware, the resistance of the tube can also lead to a gap in the measured capacitance. Measurements of the phase of the signal as well as estimates of the relative amplitudes of the real and imaginary impedances would clarify this issue.

Overall, I think this is potentially interesting work. With proper representation of its value and limits, it could be worthy of publication in Nature Communications. But in its present form it is inconsistent, misleading, overlong, and hyperbolic. I do not support publication.

Response:

In the revised manuscript:

We added data and discussion about the resistance component, and claimed it was the minor part.

The corresponding changes in the revised manuscript are listed below:

Page 11, Line 9 – Page 11, Line 15

REVIEWERS' COMMENTS:

Reviewer #1 (Remarks to the Author):

This paper describes the fabrication technique to dramatically reduce parasitic capacitance in a carbon nanotube FET structure in electrolyte solution to allow accurate evaluation of the capacitance of the carbon nanotube and the electrolyte. This measurement and theoretical modeling of the data enabled the quantification of the quantum capacitance at this interface.

Overall this revision addressed the previous problems and this paper will be of great interest for a general readership in the nanomaterial, nanoelectronic and biosensing areas.

I recommend the publication of this manuscript on Nature Communication.

Reviewer #2 (Remarks to the Author):

The authors did a complete and thorough overhaul of the paper in response to my criticisms. I find the new version much more concise, clear, and careful about what is claimed and what is not. I appreciate their efforts to undertake what was a major rewrite.

I now support publication.

One small thing, where perhaps the editor can provide guidance. A significant fraction of text from the abstract is repeated verbatim in the main body of the paper. I don't think this is appropriate, but I leave this up to the editor.

Detailed Response to Reviewer's Comments

Reviewer #1

Reviewer Comments:

This paper describes the fabrication technique to dramatically reduce parasitic capacitance in a carbon nanotube FET structure in electrolyte solution to allow accurate evaluation of the capacitance of the carbon nanotube and the electrolyte. This measurement and theoretical modeling of the data enabled the quantification of the quantum capacitance at this interface.

Overall this revision addressed the previous problems and this paper will be of great interest for a general readership in the nanomaterial, nanoelectronic and biosensing areas.

I recommend the publication of this manuscript on Nature Communication.

Response:

We would like to thank the reviewer for his/her thoughtful comments and the recognition of our work.

Reviewer #2

Reviewer Comments:

The authors did a complete and thorough overhaul of the paper in response to my criticisms. I find the new version much more concise, clear, and careful about what is claimed and what is not. I appreciate their efforts to undertake what was a major rewrite.

I now support publication.

One small thing, where perhaps the editor can provide guidance. A significant fraction of text from the abstract is repeated verbatim in the main body of the paper. I don't think this is appropriate, but I leave this up to the editor.

Response:

In the revised manuscript, we have rewritten the abstract to avoid the overlap and also to meet the word limit.

We would like to thank the reviewer for his/her important comments, which helped a lot in the improvement of our manuscript.